# Comparative Analysis of Mannans Extraction Processes from Spent Yeast *Saccharomyces cerevisiae*

**DOI:** 10.3390/foods11233753

**Published:** 2022-11-22

**Authors:** Margarida Faustino, Joana Durão, Carla F. Pereira, Ana Sofia Oliveira, Joana Odila Pereira, Ana M. Pereira, Carlos Ferreira, Manuela E. Pintado, Ana P. Carvalho

**Affiliations:** 1Escola Superior de Biotecnologia, CBQF—Centro de Biotecnologia e Química Fina—Laboratório Associado, Universidade Católica Portuguesa, Rua Diogo Botelho 1327, 4169-005 Porto, Portugal; 2Amyris Bio Products Portugal, Unipessoal Lda, Rua Diogo Botelho, 1327, 4169-005 Porto, Portugal

**Keywords:** mannans, thermal hydrolysis, autolysis, enzymatic hydrolysis, structural characterization, physicochemical characterization, cost assessment

## Abstract

Mannans are outstanding polysaccharides that have gained exponential interest over the years. These polysaccharides may be extracted from the cell wall of *Saccharomyces cerevisiae*, and recovered from the brewing or synthetic biology industries, among others. In this work, several extraction processes—physical, chemical and enzymatic—were studied, all aiming to obtain mannans from spent yeast *S. cerevisiae*. Their performance was evaluated in terms of yield, mannose content and cost. The resultant extracts were characterized in terms of their structure (FT-IR, PXRD and SEM), physicochemical properties (color, molecular weight distribution, sugars, protein, ash and water content) and thermal stability (DSC). The biological properties were assessed through the screening of prebiotic activity in *Lactobacillus plantarum* and *Bifidobacterium animalis*. The highest yield (58.82%) was achieved by using an alkaline thermal process, though the correspondent mannose content was low. The extract obtained by autolysis followed by a hydrothermal step resulted in the highest mannose content (59.19%). On the other hand, the extract obtained through the enzymatic hydrolysis displayed the highest prebiotic activity. This comparative study is expected to lay the scientific foundation for the obtention of well-characterized mannans from yeast, which will pave the way for their application in various fields.

## 1. Introduction

Large amounts of yeast cell wall are available as a by-product in the bakery, fermentation, winemaking and synthetic biology industries. Its recovery and reuse are excellent examples of a circular economy approach [1]. The low cost, non-toxicity and compositional properties of this by-product provides an opportunity for its effective conversion into value-added functional ingredients, which in turn reduces the environmental impact of the original processes.

*Saccharomyces cerevisiae* has been well characterized over the years, and is considered a valuable model for studying the cell wall of other yeasts [2]. The three main components of its cell wall are (i) glucans (1,3-β-d-glucan and 1,6-β-d-glucan), (ii) mannoproteins—mannans connected to protein—which comprise around 35 to 40% of cell dry mass [3] and (iii) chitin [3,4,5,6].

The mannoproteins can be divided into three groups: non-covalently and covalently bound to the structural glucan, and disulphide bound to other proteins that are covalently bound to the glucan of the cell wall [7]. According to Abbott et al. [8], mannans consist of a highly branched complex carbohydrate, with the α-1,6 backbone as the main chain and α-1,2- and α-1,3-linked mannose side chains. It has been reported that mannans possess several biological properties such as the inhibition of pathogen adherence, modulation of bacterial growth [9] and improvement of the immune response [10,11], thus being commonly used in animal feed as antibiotic replacers [9,12]. Additionally, the ability to scavenge radicals such as superoxide anions and hydroxyl radicals provides a potential antioxidant effect [13]. Furthermore, mannans exhibit many techno-functional properties that make them attractive for food applications derived from their physicochemical properties (water solubility, viscosity and stability); typical applications are based on their use as a hardener ingredient and emulsion stabilizer [14]. All these properties reveal the potential applications of mannans in different areas that range, but are not limited to, food [14,15], feed [14,16], cosmetics [17,18] and drug delivery [19].

The extraction of mannans usually includes the following steps: cell lysis; fractionation; and purification [20]. The step of cell lysis can be performed by different processes, namely by chemical (e.g., alkaline reagents such as NaOH [21,22] or buffer solutions such as sodium phosphate, citrate or Tris [6,7,23,24]), physical (using temperature and pressure), enzymatic (e.g., proteases [6,25,26], glucanases [6,27,28,29] and carbohydrases [7,30,31]) or mechanical processes (e.g., glass beads) [6]; a combination of processes is also widely employed. Their recovery from the solution (fractionation and/or purification) can be easily performed by precipitation—ethanol, methanol, or acetone are the solvents commonly reported in the literature for this purpose [24,32,33]. Additionally, chromatographic methods [31] and ultrafiltration [33,34] are also reported in the literature as methodologies to purify mannans. These different conditions and strategies were previously reviewed and critically compared elsewhere [20]. 

Overall, this study aimed to perform an integrative comparison of the isolation of mannans from the spent yeast *S. cerevisiae* through different processes (heat treatment in aqueous neutral and alkaline medium, autolysis and enzymatic hydrolysis with Zymolyase^®^-20T), in terms of the yield and mannose content of the obtained extracts. Structural features of all mannans extracts were unveiled by solid-state characterization techniques (ATR-FT-IR, PXRD and SEM), and their physicochemical properties (color, total protein, neutral sugars, molecular weight distribution, solubility, dry weight and ashes), along with the thermal stability, were studied. Additionally, an evaluation of the extracts’ cytotoxicity and prebiotic activities is also presented and properly compared. The structural and physiochemical characterization is highly demanded, since the functional properties of mannans are dependent on these properties. To complement the comparison of the different processes, a cost assessment was performed. This comparative study is expected to lay the scientific foundation for the recovery of well-characterized mannans from yeast, which will pave the way for their application in various fields.

## 2. Materials and Methods

### 2.1. Materials and Equipment

Spent *S. cerevisiae* was kindly provided by Amyris Company (Emeryville, CA, USA), and bacteria *Lactobacillus plantarum* and *Bifidobacterium animalis* spp. *lactis* BB12^®^ were purchased from Chr. Hansen (Hørsholm, Denmark). Zymolyase^®^-20T from *Arthrobacter luteus* was purchased from Amsbio (Cambridge, MA, USA). For reagents: Ethanol (≥99%) was purchased from CHEM-LAB (Zedelgem, Belgium); H_2_SO_4_ was purchased from Honeywell (Charlotte, NC, USA); NaOH was purchased from LabChem (Johannesburg, South Africa); and HCl was purchased from Sigma-Aldrich (St. Louis, MO, USA). All reagents were used as received without further purification. Equipment used in this study were: centrifuge, from ThermoFisher Scientific (Waltham, MA, USA); autoclave, from Prohs (Porto, Portugal); orbital system, from Eppendorf (Madrid, Spain); water bath (SW22, Julabo GmbH, Seelbach, Germany); and lyophilizer, from Christ, Osterode am Harz (model Alpha 2-4 LSCplus, Germany). 

### 2.2. Isolation and Purification of Mannoproteins

#### 2.2.1. Preparation of Spent Yeast

Spent yeast was centrifuged at 8000 rpm for 10 min, the supernatant was discarded, and the pellet was washed twice with water. 

#### 2.2.2. Thermal Hydrolysis (TH)

The process was carried out according to Freimund et al. [35], with slight modifications. In short, the yeast pellet was suspended in deionized water (10% w/v), and the pH adjusted to 7.0 (NaOH 6 M). The suspension was placed in an autoclave programmed at 120 °C for 3 h. The resulting suspension was centrifuged (5000 rpm, 10 min), and the supernatant was collected, precipitated with twice its volume of cold (4 °C) absolute ethanol and left overnight at 4 °C. The resulting precipitate was collected by centrifugation (5000 rpm, 10 min) and lyophilized.

#### 2.2.3. Thermal Hydrolysis with NaOH (TH NaOH 0.25 M and TH NaOH 1.5 M)

The process was carried out according to Liu & Huang and Lee & Ballou [22,36], with slight modifications. A yeast pellet (10% w/v) was suspended in NaOH solutions at 0.25 M and 1.5 M. The suspensions were heated to 120 °C in an autoclave for 3 h, centrifuged (5000 rpm, 10 min), and the supernatants were collected. The pH was adjusted to 7.0 (HCl 6 M), precipitated with twice its volume of cold (4 °C) absolute ethanol, and left overnight at 4 °C. The extracts were collected by centrifugation (5000 rpm, 10 min) and lyophilized.

#### 2.2.4. Autolysis (ATL) and Thermal Hydrolysis (ATL + TH)

Autolysis was performed according to Milić et al. [37] and Jacob et al. [38], with slight modifications. The yeast pellet was briefly suspended in deionized water (50% w/v), with pH adjusted to 5.2 (NaOH 10 M). The suspension was heated at 52 °C in an orbital system for 16 h under 120 rpm, and then inactivated by heating to 100 °C for 10 min in a water bath. The resulting products (pellet and supernatant) were thus separated by centrifugation (3500 rpm, 10 min). The supernatant was precipitated with ethanol as described in 2.2.2., thus obtaining the extract ATL. The pellet fraction was submitted to thermal hydrolysis treatment, as previously described. In brief, the pellet was suspended in water (10% w/v), the pH adjusted to 7.0, the pellet was heated to 120 °C in an autoclave for 3 h and the supernatant collected by centrifugation (5000 rpm, 10 min). The resulting supernatant was precipitated with cold ethanol, as previously described in the above treatments, and the extract ATL + TH was thus obtained.

#### 2.2.5. Enzymatic Hydrolysis (EH)

The enzymatic hydrolysis was performed according to Li et al. [31] and Li & Karboune [7,30], with some modifications. The yeast pellet was suspended in sodium phosphate buffer 0.5 M, pH 7.5 (5% w/v), and the enzyme was added at a concentration of 167 units of β-1,3-glucan laminaripentaohydrolase (one of the components of Zymolyase) per 1 g of yeast. The suspension was heated to 35 °C in an orbital system for 4 h under 130 rpm agitation, and then inactivated by heating to 60 °C for 10 min, in a water bath. The supernatant was collected by centrifugation (5000 rpm, 10 min) and treated as described in the previous treatments, thus allowing the EH product to be obtained.

### 2.3. Structural Characterization

#### 2.3.1. Attenuated Total Reflection Fourier-Transform Infrared Spectroscopy—ATR-FT-IR

The ATR-FT-IR analyses were performed using the Frontier™ MIR/FIR spectrometer from PerkinElmer in a scanning range of 550–4000 cm^−1^ for 16 scans at a spectral resolution of 4 cm^−1^. 

#### 2.3.2. Powder X-ray Diffraction—PXRD

The Powder X-ray Diffraction (PXRD) analyses were performed on Rigaku MiniFlex 600 diffractometer with Cu kα radiation, with a voltage of 40 kV and a current of 15 mA (3° ≤ 2θ ≥ 90°; a step of 0.01 and speed rate of 3.0°/min). 

#### 2.3.3. Scanning Electron Microscopy—SEM

The morphology of all mannans extracts was evaluated by Scanning Electron Microscopy on a Thermo Scientific™ Pro Scanning Electron Microscope. Before the analysis, the samples were placed into observation stubs covered with double-sided adhesive carbon tape (NEM tape, Nisshin, Japan), and coated with Au/Pd (target SC510-314B from ANAME, S.L., Madrid) using a Sputter Coater (Polaron, Bad Schwalbach, Germany). All observations were performed in high vacuum mode with an acceleration voltage of 5 kV. The specimen was observed using a 4000× magnification, and all images are representative of the morphology of each extract.

### 2.4. Physicochemical Properties

#### 2.4.1. Color

To determine the color point, a portable CR-410 Chroma meter (from Minolta Chroma, Osaka, Japan) was used. The CIELAB color system (L*, a*, b*) was used to determine the color point, in which the L* corresponds to the luminosity coordinate (L* value of 100 for a white object and 0 for a black object); the a* sets the green-red coordinate, and the b* sets the blue-yellow color coordinate [39]. Color was measured on the surface of the mannans samples in a petri dish placed on a white standard plate (L* = 93.22, a* = −0.08 and b* = 4.04). The total color difference (ΔE*) was calculated according to Equation (1) against a mannan commercial product from *S. cerevisiae* by Sigma-Aldrich.
ΔE* = ((L2* − L1*)^2^ + (a2* − a1*)^2^ + (b2* − b1*)^2^)^1/2^(1)

#### 2.4.2. Total Protein

For the evaluation of total protein, a BCA Protein Assay Kit (Pierce, Bonn, Germany) was used, according to the manufacturer’s instructions, in 96-well microplates (25 µL sample/200 µL BCA working reagent; 37 °C/30 min; 562 nm). A stock solution of BSA (Sigma, Munich, Germany) was prepared before carrying out the experiments. BSA dilution curves were made accordingly, and aliquots were stored to ensure an identical BSA content for each experiment. For each microplate prepared, a BSA dilution calibration curve was added, consisting of 8 points up to 1000 µg/mL BSA.

#### 2.4.3. Neutral Sugars 

The quantification of neutral sugars was performed according to what has been reported by Pinto et al. [4]. Mannans extracts were hydrolyzed using 72% H_2_SO_4_ (3 h, room temperature) followed by H_2_SO_4_ 1 M (1 h, 120 °C) [40]. Neutral sugars were derivatized to their alditol acetates, as described by Blakeney et al. [41], and analyzed by gas-chromatography-flame ionization detection (GC-FID) (Agilent Technologies, Inc., Santa Clara, CA, USA) in a 7890B GC System with a DB-225 capillary column (30 m length, 0.25 mm diameter, 0.15 µm thickness). The carrier gas was nitrogen at constant flow (1 mL/min), and hydrogen (30 mL/min) and compressed air (400 mL/min) were also used in the analysis. A split ratio of 1:60 was also used, and the injector and detector temperatures were set at 220 °C and 230 °C, respectively. The oven temperature was first set to 200 °C, and kept at that temperature for 2 min, then raised to 220 °C at 40 °C/min, with a hold time of 7 min, followed by a second linear increase to 230 °C at 20 °C/min, holding for 5 min. The total run time is 15 min. 

#### 2.4.4. Molecular Weight Distribution 

The Agilent 1260 Infinity II HPLC system was used to determine the molecular weight distribution, equipped with a vial sampler, quaternary pump, thermostatic oven and refractive index (RID) detector. Agilent Technologies’ OpenLAB CDS ChemStation was used for data acquisition and analysis. Agilent SEC, PL Aquagel-OH Mixed-M (250 × 4.6 mm, 8 µm) and PL Aquagel-OH 20 (300 × 7.5 mm, 5 µm) columns were used for separation. A calibration curve of pullulan standards was used to estimate molecular weight.

An aliquot of 10 µL of standards and test solutions was injected and eluted with the solvent (Ammonium acetate 10 mM) at a flow rate of 0.5 mL/min under isocratic conditions. The columns were kept at 50.0 °C, and the RID was set to 35.0 °C. 

#### 2.4.5. Dry Weight and Ashes

To evaluate the moisture content according to the Association of Official Analytical chemists [42], mannans samples were placed at 105 °C in a convection oven for 24 h. To determine the ash content, samples were placed at 550 °C in a muffle for 36 h for incineration. The ash was weighed after equilibration at room temperature. 

#### 2.4.6. Solubility Test

The solubility tests were performed according to the European Pharmacopoeia [43]; 1.0 g of the finely ground compound was dissolved in increasing volumes of solvent (water), and its classification, in terms of solubility, was directly related to the volume of solvent necessary for complete solubilization. Substances that dissolve in 10–30 mL, 30–100 mL, 100–1000 mL or 1000–10,000 mL of solvent are classified as soluble, sparingly soluble, slightly soluble or very slightly soluble, respectively. If the substance does not dissolve, it is classified as insoluble in this solvent.

#### 2.4.7. Differential Scanning Calorimetry—DSC 

Differential scanning calorimetry (DSC) measurements were performed under a nitrogen atmosphere using DSC 204 F1 Phoenix equipment from Netzsch, calibrated using an indium standard. The samples (3–6 mg) were placed into aluminium DSC pans with a pinhole, and an empty pan was used as a reference. Heating from 20–300 °C was carried out at a heating rate of 10 °C/min.

### 2.5. Cytotoxicity Evaluation

#### 2.5.1. Cell Line Growth Conditions

Human colon carcinoma (Caco-2) cells were obtained from the European Collection of Authenticated Cell Cultures. They were grown using high glucose (4.5 g/L) Dulbecco’s Modified Eagle Medium (DMEM) supplemented with 10% (v/v) heat-inactivated Fetal Bovine Serum (FBS), 1% (v/v) antibiotic and antimycotic 100x (Gibco, Milan, Italy) and 1% (v/v) of non-essential amino acids 100x (Sigma, Germany). Cells were used between passages 55 and 56.

#### 2.5.2. Cytotoxicity Assay

Cytotoxicity of the samples was assessed in the CaCo-2 cell line in conformity with ISO 10993-5 [44], using the PrestoBlue™ Cell Viability Reagent (Thermo Fisher Scientific, Waltham, MA, USA), according to the instructions of the manufacturer. The cells in suspension were seeded at 1 × 10^4^ cells/well in a 96-well microtiter plate. After 24 h, culture medium from the confluent cells was removed and replaced with the samples. Samples were previously dissolved in phosphate buffer saline (PBS) solution, pH 7.4, to a final concentration 2-fold higher than the desired, and twofold diluted in antibiotic containing DMEM in the range of 10.0–0.31 mg/mL. After 24 h of incubation, the PrestoBlue (PB) reagent was added to the wells, and changes in cell viability were detected using fluorescence spectroscopy (excitation 570 nm; emission 610 nm).

### 2.6. Screening of Prebiotic Effect

The potential prebiotic effect of the mannans extracts from the different extraction methods was determined by screening their impact upon the growth of two potential probiotic *Lactobacillus plantarum* and *Bifidobacterium animalis* spp. *lactis* BB12^®^. All strains were used as monoculture, and were grown in MRS broth at 37 °C for 24 h, under aerobic conditions for *L. plantarum* and anaerobic conditions for *B. animalis* BB12. The mannans extracts and fructo-oligosaccharides (FOS) (Sigma-Aldrich Chemistry, St. Louis, MO, USA), were added to sterilized MRS broth without glucose to a final concentration of 2% (w/v). MRS broth with glucose (2%) was used as negative control (Control). From each MRS medium inoculated with *L. plantarum* and *B. animalis* subsp *lactis* BB12, 200 µL were transferred, in triplicate, to a 96-well microplate (Nunc, Denmark). Additionally, 50 µL of paraffin was added to ensure anaerobic conditions in the case of the *B. animalis* subsp *lactis* BB12. The microplates were incubated for 48 h at 37 °C, and absorbance was measured at 625 nm each hour with a multidetector plate reader (Epoch, VT, USA).

### 2.7. Cost Assessment

The costs associated with the different processes were estimated considering two major components: energy and reagent consumption. Both parameters were calculated at laboratory scale, considering the average yields and consumptions of the triplicate experiments performed for each extraction process. The electricity price used for calculation purposes was 0.2067 €/kWh, which was the average household price in Portugal in 2022 [45]. Energy consumption was estimated by considering the maximum power of each equipment and the time that same equipment was used. The water price used in the calculations was 1.8198 €/m^3^ [46]. 

### 2.8. Statistical Analysis

All extractions and analyses were performed in triplicate (n = 3). Extraction yield, protein quantification, total sugar and glucose, dry weight, ashes content and main population distribution means were compared using one-way ANOVA followed by Tukey’s Multiple Comparison Test, using a 95% confidence interval as criteria. The normality of the samples was evaluated using the Shapiro-Wilk’s Test. All assays were performed using the Statistical Package for Social Sciences software (version 21, SPSS, Chicago, IL, USA).

## 3. Results and Discussion 

Several studies indicate that the cell wall and cytoplasm of yeast cells contain many nutritional components [47,48]. To use its components effectively, however, it is necessary to overcome the difficulty of breaking through the thick and rigid cellular walls [49] of the yeast. Various methods/processes for cell wall rupture have been developed through chemical, biological, physical and mechanical approaches. 

In this study, after a review of the most used conditions to extract mannans from *S. cerevisiae* yeast in the literature, some particular conditions of physical (temperature and pressure), chemical (alkaline and neutral solvents) and enzymatic (autolysis and enzymatic hydrolysis) extraction processes were selected. The efficiency of each of these extraction methods in terms of yield (considering the whole extract—solid yield, or just the mannose fraction—mannose yield) and mannose content were evaluated, as described in Table 1.

The highest production yield (mannose yield, 58.82%) was achieved by thermal hydrolysis with 0.25 M of NaOH (TH NaOH 0.25 M), though the corresponding mannose content was low (32.91%). On the other hand, autolysis followed by a hydrothermal step (ATL + TH) allowed the highest mannose content to be reached (59.19%). Thermal hydrolysis in neutral conditions (TH) presents the second-highest mannose content (53.46%), but more than twice the yield obtained for ATL + TH. The use of enzymes poses a significant economic burden [20], and in this particular process, and looking to the parameters of Table 1, its use may not be justified, as the yield and content values obtained were not as appealing as with other processes. On the other hand, the use of high concentrations of NaOH in processes TH NaOH 0.25 M and TH NaOH 1.5 M results in extracts with a high level of salt (NaCl), as discussed subsequently, which may imply additional steps of downstream processing.

### 3.1. Structural Analysis 

The structural analysis of the mannans extracts from *Saccharomyces cerevisiae* obtained by different extraction methods was attained by the following set of solid-state techniques: Attenuated Total Reflection Fourier-transform Infrared Spectroscopy (ATR-FT-IR); Powder X-ray Diffraction (PXRD); and Scanning Electron Microscopy (SEM). The normalized ATR-FT-IR spectra of all products are depicted in Figure 1. 

The analysis of the spectra highlighted in Figure 1 allows inferring that all mannans extracts share similar spectral properties. All spectra exhibit a strong and broad vibration band at 3000–3687 cm^−1^, corresponding to the stretching of the hydroxyl groups [13,50]; a vibration at ca. 2943 cm^−1^, which can be attributed to the C-H stretching vibration; and a vibration at 1403 cm^−1^, which can be assigned to the to the C-H bending vibration. All ATR-FT-IR spectra also exhibit a sharp band at 1647 cm^−1^, which can be attributed to the C-O asymmetric stretching vibration [51]; a vibration band at 1025 cm^−1^ assigned to the O-H variable angle vibrations and the characteristic absorption of the mannan α-chain appeared at 814 cm^−1^ [13,50]. The spectra corresponding to the mannan extracts obtained by autolysis and enzymatic hydrolysis, ATL and EH, respectively, exhibit two strong vibrations at 2925 and 2854 cm^−1^ (lipid region) that can be assigned to the asymmetric and symmetric vibrational modes of CH_2_ groups, respectively, and a sharp vibration at 1743 cm^−1^ can be attributed to the C=O stretching, generally representing lipids [52].

The analysis of the crystallinity of the mannans extracts obtained from the various extraction methods was assessed by Powder X-ray Diffraction (PXRD), as depicted in Figure 2.

The PXRD patterns of the mannans extracts obtained by pH neutral hydrolysis (TH), autolysis (ATL) or even the product resulting from the combination of both previous methods (ATL + TH), along with the extract obtained by enzymatic hydrolysis (EH), exhibit a broad diffraction at 2θ = 19.7°, which is consistent with the predominant amorphous character (Figure 2a) [53]. However, the PXRD analysis of the products resulting from the alkaline thermal hydrolysis (Figure 2b), using different NaOH concentrations of 0.25 and 1.5 M, TH NaOH 0.25 M and TH NaOH 1.5 M, respectively, revealed the presence of NaCl—produced in the neutralization reaction of NaOH and HCl—with both samples exhibiting the characteristic diffraction peaks of NaCl (27.5°, 31.7°, 45.4°, 53.7°, 56.6°, 66.2°, 75.5°and 84.1°) [54]. These extracts, resulting from the alkaline extraction processes, require further purification steps in order to prompt the salt removal, namely by further washing with water, or even by desalination. 

The morphology of all mannans extracts was evaluated by Scanning Electron Microscopy (SEM), as highlighted in Figure 3. 

The study of morphology/size is highly relevant, since these properties may affect physicochemical parameters, namely wettability and solubility. The TH and EH extracts are constituted by rough spherical particles. However, while the particles corresponding to the extract obtained by thermal hydrolysis are the smallest, the EH extract exhibits large aggregates. The product resulting from the autolysis process (ATL) was revealed to be constituted of small aggregates of plates and rough spherical particles, and the extract obtained by the combination of extraction processes (ATL + TH) showed the most heterogeneous nature, constituted by large smooth plates and large aggregates of small plates and rough spherical particles. The products obtained from the alkaline hydrolysis processes are characterized by large plates coated with rough spherical particles and large plates with spherical particles for TH NaOH 0.25 M and TH NaOH 1.5 M, respectively.

### 3.2. Physicochemical Characterization 

The integrated and comparative analysis of the physical appearance of all mannans products (TH, TH NaOH 0.25 M, TH NaOH 1.5 M, ATL, ATL + TH and EH), along with their physicochemical properties and thermal analysis, is presented in this section.

#### 3.2.1. Physical Appearance 

The physical aspect of the powders is significantly different, as shown in Table 2. All the mannans extracts appeared as a homogeneous fine powder. The mannans extracts of TH NaOH 0.25 M and TH NaOH 1.5 M appeared as a gold tone. The extracts ATL and EH are brownish, while the TH and ATL + TH extracts revealed an off-white color. The color of products constitutes an important criterion for their commercialization, as the colors accepted by the consumers are very scarce. The color point of all mannans extracts was measured using the system CIELAB (L*, a*, b*) [39]. 

The TH extract exhibits the highest L* (lightness) value (83.88), followed by the TH NaOH 1.5 M (81.06) and TH NaOH 0.25 M (76.92), ATL (76.36) and ATL + TH (76.74), respectively, and the EH (74.81) reveals the lowest L* value. 

The total color difference (ΔE*) was determined to allow the quantification of the color change between the yeast mannans products after the various extracting processes and a commercial mannan product of *S. cerevisiae* by Sigma-Aldrich. The ΔE* (1.50 < ΔE* > 3.0) between the differences extracts and the benchmark was 6.43 for TH extract, 15.10 for TH NaOH 0.25 M extract, 10.76 for TH NaOH 1.5 M, 14.35 for ATL, 13.45 for ATL + TH and finally 15.81 for EH. Since the determined total color differences were higher than 3.0, this corroborates the distinct visual perception among all mannans samples [55]. 

#### 3.2.2. Physicochemical Properties

The physicochemical analyses performed for the mannans (TH, TH NaOH 0.25 M, TH NaOH 1.5 M, ATL, ATL + TH and EH) extracts are summarized in Table 3. All mannans extracts were evaluated regarding their composition—protein and total sugars (mannose and glucose)—and molecular weight distribution (evaluated by HP-SEC). The moisture and ash content, along with solubility, were also studied for all products.

The yeast cell wall is made up of about 85–90% polysaccharides and 10–15% protein [56]. In the cell wall, most of the proteins are covalently linked to mannans, forming mannoproteins, which are in the outer layer of the wall and have ‘gel-like’ properties to protect the yeast cells. 

To break down the yeast cell wall to extract the mannans, different extraction methods were used, such as alkaline methods (0.25 M and 1.5 M NaOH), neutral and enzymatic (autolysis and enzymatic hydrolysis), and combining with physical methods (pressure, time and high temperature). In theory, by breaking down the cell wall of yeast, mannoproteins and mannans are released into the aqueous medium, which can be separated from the remaining insoluble components of the cell wall, such as glucans, by centrifugation. This is followed by precipitation with absolute ethanol at low temperature, which will allow the separation of mannans and mannoproteins from other soluble components present in the supernatant. These extraction methods, followed by absolute ethanol purification, were not found to be sufficiently effective to eliminate the bound protein. Initial reports on mannans extraction from yeast cell walls used a thermal treatment under alkaline conditions [21,57]. Later studies revealed that heat treatment at neutral pH obtained similar extracts [32], with the advantage of reducing the number of steps required in the process. Both alkaline extraction processes presented the best results in term of the elimination of proteins from the mannoprotein complex. The glycosyl-serine and glycosyl-threonine bonds, phosphodiester bonds, some peptide bonds and disulphide and acyl ester bonds may be broken in these conditions, particularly in the most alkaline environment [58]. Even though the extraction using water as a solvent (TH) potentially generates material with intact phosphodiester bonds, and presumably intact peptide bonds, the protein is likely denatured [59]. The processes that result in the highest protein content are enzymatic hydrolysis (EH) and autolysis (ATL), with 21.67% and 19.33%, respectively. Their mild temperature conditions are favorable to the preservation of proteins.

Regarding the sugar content obtained by the different methods (Table 3), ATL + TH extract had the highest total sugar content (65.75% w/w), while TH NaOH 1.5 M extract had the smallest total sugar content (24.21% w/w), which also corresponds to the highest and lowest mannose content extracts, 59.19% w/w and 19.67% w/w, respectively (Table 1). In the ATL + TH extraction process, initial autolysis releases part of the mannoproteins in the cell wall into the aqueous medium, which resulted, following the precipitation with ethanol, in a mannose content of 25.14% w/w (ATL). The subsequent extraction from the remaining pellet resulted in a significantly purified extract. This double-step methodology may be further explored to obtain higher purity extracts.

MW has been reported to have a direct relationship with bioactivity [60]. Molecular weight distribution analysis revealed significant differences not only in the MW, but also in the number of populations, obtained by the different extraction processes, as shown in Table 3. The MW of the most significant population, evaluated in terms of % area, is significantly decreased in alkaline methods. According to Liu et al. [13], the alkaline environment would hydrolyze mannans to populations with smaller molecular weight, which is in accordance with our analysis where extracts from TH NaOH 0.25 M and 1.5 M present not only a reduced MW on the highest MW population, at around 192 kDa (versus 214 kDa to 237 kDa in the other extraction processes), but also a reduced %area, with 29 ± 1% and 10 ± 1%, respectively. The reduction may be due to the hydrolysis of mannans, as mentioned by Liu et al. [13], but also as a result of the loss of protein bound to mannans in these extracts. A study by Galinari et al. [61] reported that mannans extracted from fungus (*Kluyveromyces marxianus*) presented 5 different fractions with MW from 10–203 kDa. Although the extraction processes are different, they reveal the same two populations with a molecular weight ranging from 190–240 kDa populations. These are the only populations present in extracts from TH and ATL + TH, but the remaining extracts present a larger variety of populations with high significance in the range of 4–240 kDa. The highest value is slightly larger than those reported in the literature, which can be due to the origin and genetic modification of the yeast used. 

A significant difference was found between the extracts concerning ash content, as shown in Table 3. Extracts TH NaOH 0.25 M and TH NaOH 1.5 M showed the highest percentage of minerals (46.14 and 63.96% w/w, respectively). This scenario was already expected, since the sodium chloride by-product resulting from the neutralization process (by-product confirmation by PXRD analyses—Figure 3) prompts the increase of the mineral content. As previously mentioned, the higher level of impurities may require extra purification processes, such as desalination, to meet the desired purity requirements.

Regarding the solubility of the different extracts, TH NaOH 0.25 M, ATL and EH extracts proved to be insoluble, which is not desirable, as it may pose difficulties in its use/incorporation into new products. In addition, the latter method showed other disadvantages, such as the cost of using enzymes, which may not pay off (considering the low mannose content values). The low solubility of these extracts (TH NaOH 0.25 M, ATL and EH) may be an important limitation for different applications. 

The extract TH NaOH 1.5 M showed the highest solubility, at approx. 5 mg/mL, which may be due to the high salt content, which may facilitate its solubility. However, its mannose content (19.67% w/w) is low when compared to the extracts obtained by other processes. 

#### 3.2.3. Thermal Analysis

The thermal properties of the mannans extracts (TH, TH NaOH 0.25 M, TH NaOH 1.5 M, ATL, ATL + TH and EH) were studied by differential scanning calorimetry (DSC), as depicted in Figure 4.

The DSC thermograms were very consistent with each other, though some slight differences were observed. The first event, which was observed for all extracts, was an endothermic event at around 70 to 80 °C, which was attributed to the evaporation of the water bound to the polysaccharides present in the sample [62,63]. This endothermic event was observed in all extracts. 

Only the TH, ATL + TH and ATL reveal an exothermic event of around 260–290 °C; this peak can be related to polysaccharide thermal decomposition. According to Ospina Álvarez et al. [64], events close to or above 300 °C are typically observed during the degradation of the saccharide structure, and may be involved in the dehydration of saccharide rings. Another study in fungi reveals that the exothermic event at around 300 °C, more specifically at 334 °C, refers to the oxidation of the mannans [65].

Another possibility to explain the position of the exothermic peak at around 300 °C in TH, ATL + TH and ATL extracts was the presence of chitosan, a deacetylated derivative of chitin [66]. The cell wall of spent yeast contains 1–2% of chitin [3,67] and chitin bonded to β-(1-3)-d-glucan [68]. Consequently, some residues of chitin bonded β-glucan may be present in the mannans extracts after the extraction of the spent yeast.

The presence of the β-glucan on the mannans extracts is very possible, because as shown above, there is a percentage of glucose in the extracts which we can associate with the presence of β-glucan residues. The extracts ATL + TH and ATL presented a well-defined peak against the TH extract, and according to the content of the glucose, the extracts ATL + TH and ATL expose a higher content (6.56 and 14.81%, respectively), which means there is a possibility that this peak is not just degradation but the presence of β-glucan, and in turn, β-glucan linked to chitin.

However, extracts such as TH NaOH 0.25 M, TH NaOH 1.5 M and EH reveal curves that indicate remarkable thermal stability, as they do not present this degradation or impurity peak.

#### 3.2.4. Cytotoxicity

Cytotoxicity of samples was assayed against CaCo-2 cells, by evaluating their impact on cell metabolism using a resazurin-based dye as cell viability indicator (Figure 5). According to ISO 10993-5 [44], a sample is cytotoxic when a metabolic inhibition percentage of above 30% is observed.

All mannans extracts (TH, TH NaOH 0.25 M, TH NaOH 1.5 M, ATL, ATL + TH and EH) exhibited no cytotoxicity up to 10 mg/mL. This indicates that the differences in the structural and physicochemical properties of the different products do not negatively impact their interactions with the CaCo-2 cells. Moreover, the highest metabolic stimuli were found in the products resulting from the neutral pH extract, TH, (−82.1%) and TH NaOH 0.25 M (−79.1%).

#### 3.2.5. Screening of Prebiotic Effect

Nowadays, the prebiotic concept has expanded, in part due to the advances in the tools for microbiome research, which have improved our knowledge of the composition of the microbiota, and enabled the identification of additional substances influencing colonization [69]. To evaluate the prebiotic potential of mannans extracts, growth curves, obtained by optical density (OD), of the two probiotic strains *L. plantarum* and *B. animalis* subsp *lactis* BB12 were performed in microplates with MRS without glucose media, supplemented with the mannans extracts resulting from the different extraction processes. The 2% concentration of the carbon source was chosen based on the composition of the commercial MRS medium containing 2% (w/v) glucose. The FOS was selected to compare with the tested mannans extracts, since it is a standard prebiotic.

Analysis of the growth curves controls for *L. plantarum* controls (Figure 6a) revealed that the bacteria presented a reasonable growth rate in basal medium MRS with glucose (negative control) (1.081 ± 0.104), and a boosting growth when MRS medium was supplemented with FOS (positive control) (1.349 ± 0.024). Regarding the prebiotic potential of the mannans extracts, it was observed that the EH extract enabled the highest increase of OD absorbance over time, reaching a maximum value of 1.370 ± 0.024 at 48 h incubation, which was higher than the positive control (FOS). An additional interesting result was obtained for the extract from the neutral pH (TH), with the growth promotion of the bacterium to an OD value of 1.328 ± 0.050.

Similar behavior was reported for *B. animalis* BB12 (Figure 6b) in control in basal MRS medium supplemented with FOS revealing the highest growth (1.484 ± 0.010). When the prebiotic potential of the mannans extracts was assessed once again, the enzymatic hydrolysis extract (EH) revealed the highest increase in OD absorbance (1.514 ± 0.039). Alternatively, it is possible to observe that extracts TH NaOH 0.25 M (0.808 ± 0.189) and TH NaOH 1.5 M (0.911 ± 0.003) affect the growth of the probiotic strains, possibly due to the high concentration of salt in these extracts.

It was reported by Madadi et al. [70] that extracts obtained from yeast, particularly α-mannans, significantly increase the amount of lactic acid bacteria (LAB) in the host. Another work by Everard et al. [71] illustrates how the α-mannans obtained from *S. boulardii* increase the proportion of *Lactobacillus* sp. in the intestinal microbiota, and have a beneficial effect on host metabolism. A study by Tang et al. [72] reported the growth of 12 pure LAB cultures in the presence of three yeast α-mannans extracts, and the obtained results suggested that yeast α-mannans could be used as a new prebiotic in functional foods to improve the intestinal environment.

### 3.3. Cost Assessment 

Mannans extraction using the numerous methodologies presented within this study revealed how the different approaches resulted in such distinctive extracts, varying not only in composition, molecular weight, solubility, physical appearance and morphology, but also in its biological performance. An additional and very relevant parameter is the assessment of the cost of each process. 

In this estimate, the energy and reagent consumption were considered the major players contributing to the cost of the process. Both parameters were calculated at laboratory scale, considering the average yields and consumptions of the triplicate experiments performed for each extraction process. Reagents were ACS grade, and the maximum power of each equipment was used. The analysis considered local energy and water prices (0.2067 €/kWh and 1.8198 €/m^3^, respectively). 

Since this assessment is performed with the goal of determining the most cost-effective process, the freeze-drying step was not taken into account, since this was the drying technology used for all extracts produced.

The results of cost, energy consumption and water consumption are presented in Table 4, both in relation to the total amount of extracts and the corresponding mannose fraction. 

The cost assessment presented in Table 4 revealed that the process with the lowest cost was autolysis (ATL), considering both the cost per gram of extract and per gram of mannose. This is essentially due to its lower water and ethanol consumption, since the initial pellet was suspended at 50% w/v, but also due to the lower energy consumption, as the process does not require such a high extraction temperature as the other procedures. This process is followed closely by the extract of TH NaOH 0.25 M, which owes its low cost to the high solid yield. Both these extracts are practically insoluble, which in some instances may limit its application.

ATL + TH is a double step process, with the sequential extraction via ATL and TH. This process presents the highest cost, as expected. It includes the high energy costs associated with the use of high temperature (autoclave) and the high water and ethanol consumption linked to the second step of the process (TH).

The EH cost was relatively high, which is a direct consequence of the price associated with Zymolayse. The immobilization of the enzyme could allow its reutilization, and the costs could be significantly reduced. In fact, the energy consumption of EH is the lowest, supporting the potential of this process for optimization. Even though the EH extract was determined to be practically insoluble, it showed great potential prebiotic activity. 

Energy and water consumption (also called energy and water intensity, when normalized to the obtained product weight) are crucial factors in the evaluation of a process sustainability. 

EH presents the lowest energy consumption, which is a consequence of the mild temperature conditions required for enzymatic hydrolysis to occur. Its water consumption was not particularly high. The fact that the extract resulting from this process was also bioactive shows the potential of enzymatic hydrolysis, if the costs of enzyme can be circumvented by methods such as enzyme immobilization. 

On the other hand, ATL presented the lowest water consumption and the lowest cost, which is also very appealing. One parameter that was not considered in the analysis is the time of processes, which in this case is particularly long. Additionally, the procedure used a pellet mass to volume ratio that was significantly different from the remaining procedures (50% w/v), which was a value based on literature. Ideally, the comparison should be performed in the same starting conditions. Nonetheless, it is an interesting and promising result. 

If we evaluate the sustainability of the process considering only these two parameters (water and energy consumption), TH NaOH 0.25 M presents good scores. However, the presence of salt in the extracts would likely require additional purification steps. 

The estimation of the costs associated with the process is fundamental in decision making, particularly when planning the scaling up of processes. Additionally, the design of sustainable processes is paramount nowadays, and it is recommended to perform this analysis early on.

## 4. Conclusions

The present study reports the effect of different extraction methods on the structural, physicochemical and biological characteristics of mannans. Additionally, the process yields and costs were evaluated. Interestingly, each extraction process was found to result in distinct characteristics, yields and costs. 

The highest solid yield was obtained by enzymatic hydrolysis, which also presented the best prebiotic performance. Its cost is significantly high due to the cost of enzymes, but this could potentially be overcome with the use of immobilized enzymes. The most cost-effective extraction process was found to be the autolysis, essentially as a result of the lower water and ethanol requirement, but also due to the mild extraction conditions which require less energy. Alkaline hydrolysis resulted in high yields, particularly using 0.25 M NaOH, although extracts become contaminated with NaCl, and may require additional purification steps depending on the purity requirements. TH is a good compromise, since it results in a high-purity extract without a major cost imposition. Potential improvement could result from increasing the mass to volume ratio, allowing the reduction of costs, energy and water consumption.

This study provides key information to help design an extraction process that fits the requirements of the intended application.

## Figures and Tables

**Figure 1 foods-11-03753-f001:**
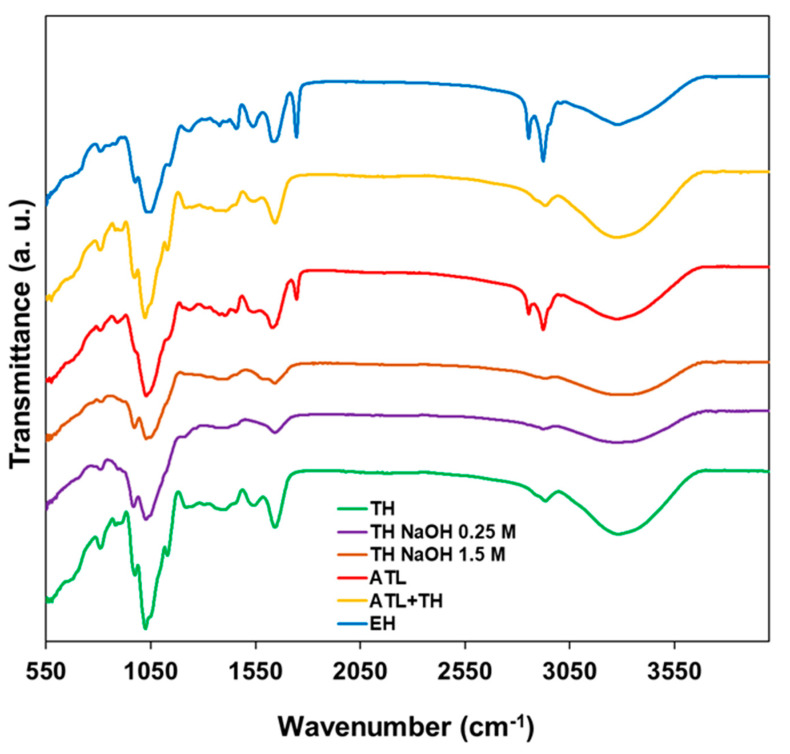
ATR-FT-IR spectra of the mannans extracts correspondent to the following extraction methods: pH-neutral Thermal Hydrolysis (TH); alkaline Thermal Hydrolysis (TH NaOH 0.25 M and TH NaOH 1.5 M); Autolysis (ATL); Autolysis followed by pH neutral Thermal Hydrolysis (ATL + TH); and Enzymatic Hydrolysis (EH).

**Figure 2 foods-11-03753-f002:**
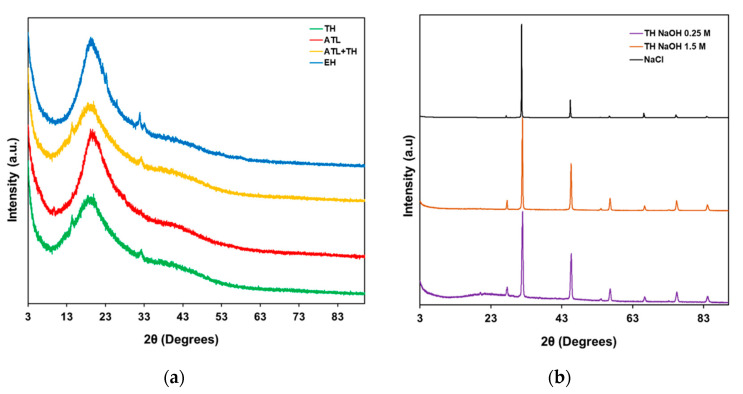
Powder X-ray Diffraction analyses of the mannans extracts correspondent to the following extraction methods: (**a**) pH neutral Thermal Hydrolysis (TH), Autolysis (ATL), Autolysis followed by pH-neutral Thermal Hydrolysis (ATL + TH) and Enzymatic Hydrolysis (EH); and (**b**) alkaline Thermal Hydrolysis (TH NaOH 0.25 M and TH NaOH 1.5 M) and commercial NaCl.

**Figure 3 foods-11-03753-f003:**
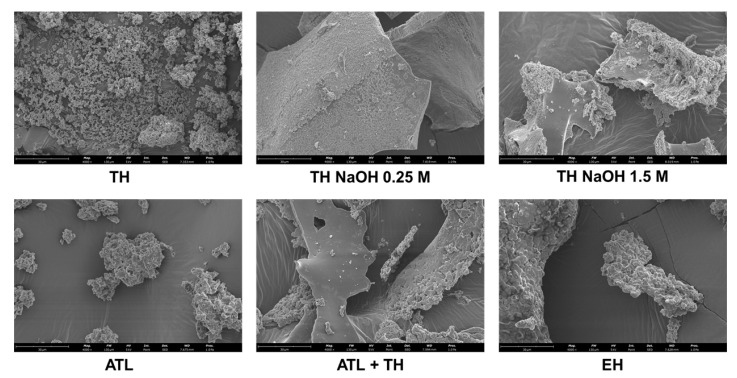
Scanning Electron Microscopy Analyses of the mannans extracts correspondent to the following extraction methods: Thermal Hydrolysis (TH); alkaline Thermal Hydrolysis (TH NaOH 0.25 M and TH NaOH 1.5 M); Autolysis (ATL); Autolysis followed by pH-neutral Thermal Hydrolysis (ATL + TH); and Enzymatic Hydrolysis (EH).

**Figure 4 foods-11-03753-f004:**
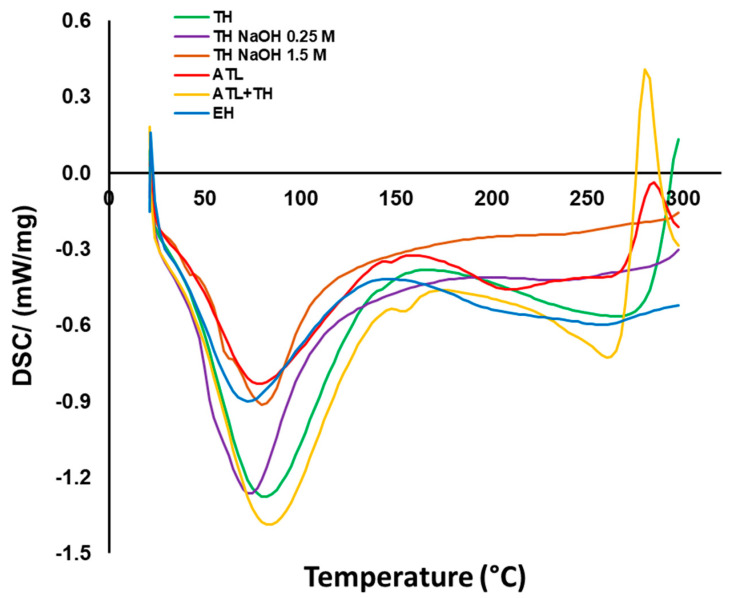
DSC curves resulting from heating the samples from 20 °C to 300 °C at a heating rate of 10 °C/min.

**Figure 5 foods-11-03753-f005:**
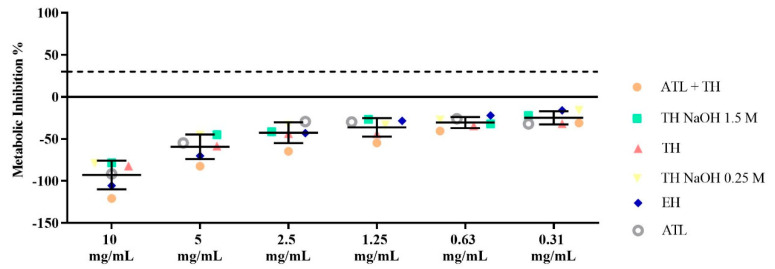
Results of the metabolic inhibition in % for the following concentrations: 10, 5, 2.5, 1.25, 0.63 and 0.31 mg/mL of the TH, TH NaOH 0.25 M, TH NaOH 1.5 M, ATL, ATL + TH and EH mannans extracts.

**Figure 6 foods-11-03753-f006:**
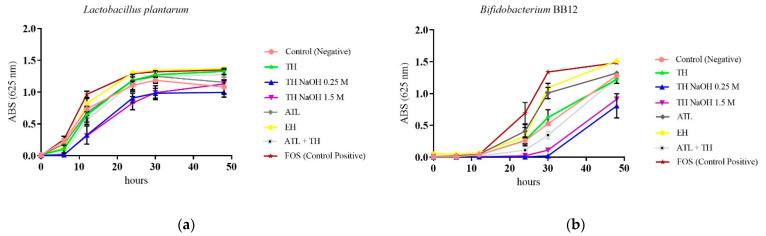
Growth curves of *Lactobacillus plantarum* (**a**) and *Bifidobacterium animalis* BB12 (**b**) in MRS broth supplemented with FOS, glucose or with mannans TH, TH NaOH 0.25 M, TH NaOH 1.5 M, ATL, ATL + TH and EH extracts, over 48 h incubation at 37 °C.

**Table 1 foods-11-03753-t001:** Solid and mannose yields and mannose content of mannan extracts obtained by different extraction methods (TH, TH NaOH 0.25 M, TH NaOH 1.5 M, ATL, ATL + TH and EH).

	TH	TH NaOH 0.25 M	TH NaOH 1.5 M	ATL	ATL + TH	EH
**Solid Yield** (**%**)	8.25 ± 0.22 ^c^	20.22 ± 1.23 ^ab^	16.66 ± 3.43 ^b^	10.18 ± 1.04 ^c^	3.58 ± 1.30 ^d^	23.88 ± 1.16 ^a^
**Mannose Yield** (**%**)	39.04 ± 0.76 ^b^	58.82 ± 1.57 ^a^	28.65 ± 3.05 ^c^	22.65 ± 2.66 ^cd^	18.75 ± 6.84 ^d^	45.80 ± 2.16 ^b^
**Mannose Content** (**%**)	53.46 ± 0.44 ^b^	32.91 ± 1.64 ^c^	19.67 ± 1.82 ^e^	25.14 ± 1.72 ^d^	59.19 ± 1.35 ^a^	21.72 ± 2.02 ^de^

Superscript different letters in the same row represent statistically different values (*p* < 0.05). TH—Thermal Hydrolysis (H_2_O), TH NaOH 0.25 M—Thermal Hydrolysis NaOH 0.25 M, TH NaOH 1.5 M—Thermal Hydrolysis NaOH 1.5 M, ATL—Autolysis, ATL + TH—Autolysis + Thermal Hydrolysis and EH—Enzymatic Hydrolysis.

**Table 2 foods-11-03753-t002:** Results of the physical appearance and color characteristics of the TH, TH NaOH 0.25 M, TH NaOH 1.5 M, ATL, ATL + TH and EH extracts.

	TH	TH NaOH 0.25 M	TH NaOH 1.5 M	ATL	ATL + TH	EH
**Physical** **Appearance**	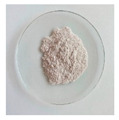	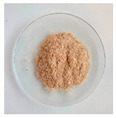	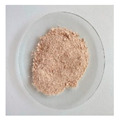	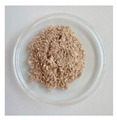	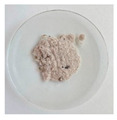	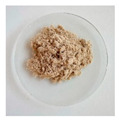
**Color** **Characteristics**						
**L***	83.88 ± 0.01 ^a^	76.92 ± 0.01 ^c^	81.06 ± 0.01 ^b^	76.36 ± 0.01 ^f^	76.74 ± 0.01 ^e^	74.81 ± 0.02 ^d^
**a***	1.36 ± 0.01 ^f^	3.68 ± 0.01 ^a^	3.22 ± 0.00 ^b^	1.72 ± 0.01 ^c^	1.46 ± 0.01 ^e^	1.69 ± 0.01 ^d^
**b***	7.08 ± 0.01 ^f^	14.03 ± 0.02 ^a^	12.48 ± 0.01 ^b^	11.45 ± 0.02 ^c^	7.64 ± 0.01 ^e^	11.37 ± 0.02 ^d^
**ΔE* ^1^**	6.43 ± 0.01 ^f^	15.10 ± 0.02 ^b^	10.76 ± 0.00 ^e^	14.35 ± 0.02 ^c^	13.45 ± 0.01 ^d^	15.81 ± 0.03 ^a^

L* is the lightness coordinate, a* defines the green-red coordinate, and b* the blue-yellow coordinate. Superscript different letters in the same row represent statistically different values (*p* < 0.05). ^1^ The calculation of ΔE* was performed against a commercial mannan product to make the comparison.

**Table 3 foods-11-03753-t003:** Results of the physicochemical characterization of mannans extracts obtained by different extraction methods: Thermal Hydrolysis (TH); alkaline Thermal Hydrolysis (TH NaOH 0.25 M and TH NaOH 1.5 M); Autolysis (ATL); Autolysis followed by pH neutral Thermal Hydrolysis (ATL + TH); and Enzymatic Hydrolysis (EH).

	TH		TH NaOH 0.25 M		TH NaOH 1.5 M		ATL		ATL + TH		EH	
**Protein** **Content** **(% w/w)**	13.00 ± 1.00 ^b^		1.67 ± 0.58 ^d^		0.33 ± 0.58 ^d^		19.33 ± 2.08 ^a^		7.00 ± 1.00 ^c^		21.67 ± 2.08 ^a^	
**Total Sugars** **(% w/w) ***	58.28 ± 0.79 ^b^		36.65 ± 2.20 ^c^		24.21 ± 2.32 ^d^		39.94 ± 2.05 ^c^		65.75 ± 2.30 ^a^		27.76 ± 2.77 ^d^	
**Glucose** **(% w/w)**	4.82 ± 0.44 ^cb^		3.73 ± 0.63 ^c^		4.54 ± 0.57 ^cb^		14.81 ± 1.43 ^a^		6.56 ± 1.24 ^b^		6.04 ± 0.75 ^b^	
**Molecular Weight (kDa)–Most** **Significant Populations**	MW (kDa)	Area%	MW (kDa)	Area%	MW (kDa)	Area%	MW (kDa)	Area%	MW (kDa)	Area%	MW (kDa)	Area%
	237 ± 1 ^a^	82 ± 2	192 ± 0 ^e^	29 ± 1	192 ± 0 ^e^	10 ± 1	221 ± 1 ^c^	65 ± 1	231 ± 5 ^b^	84 ± 1	214 ± 3 ^d^	67 ± 5
	MW (kDa)	Area%	MW (kDa)	Area%	MW (kDa)	Area%	MW (kDa)	Area%	MW (kDa)	Area%	MW (kDa)	Area%
	129 ± 2 ^c^	18 ± 2	135 ± 4 ^cb^	23 ± 9	144 ± 1 ^a^	8 ± 1	132 ± 0 ^cd^	18 ± 1	133 ± 3 ^cb^	16 ±1	136 ± 2 ^b^	16 ± 4
**Dry Weight** **(% w/w)**	5.96 ± 1.44 ^ab^		5.39 ± 2.00 ^ab^		3.79 ± 1.15 ^cb^		1.89 ± 0.85 ^cb^		8.62 ± 2.41 ^a^		0.57 ± 1.03 ^c^	
**Ashes** **(% w/w)**	6.11 ± 0.33 ^d^		46.14 ± 2.13 ^b^		63.96 ± 0.27 ^a^		5.10 ± 0.22 ^e^		3.79 ± 0.25 ^ed^		10.93 ± 0.50 ^c^	
**Solubility**	Very slightly soluble (approx. 1 mg/mL)		Practically insoluble		Slightly soluble (approx. 5 mg/mL)		Practically insoluble		Very slightly soluble (approx. 1 mg/mL)		Practically insoluble	

* Total sugars are the sum of the glucose content (% w/w) plus the mannose content (% w/w, in Table 1). Superscript different letters in the same row represent statistically different values (*p* < 0.05).

**Table 4 foods-11-03753-t004:** Results of the Cost (€), Energy (kWh) and Water Consumption (L) per gram of extract and per gram of mannose of each mannans extract process.

	Cost		Energy Consumption		Water Consumption	
	(€/g of Extract)	(€/g of Mannose)	(kWh/g of Extract)	(kWh/g of Mannose)	(L/g of Extract)	(L/g of Mannose)
**TH**	15	24	32	51	0.4	0.7
**TH NaOH 0.25 M**	7	6	13	13	0.2	0.2
**TH NaOH 1.5 M**	8	9	16	17	0.2	0.2
**ATL**	2	5	15	33	0.1	0.2
**ATL + TH**	49	67	141	194	1.4	1.9
**EH**	31	103	4	13	0.3	0.1

## Data Availability

The data presented in this study are available within the article.

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
