# Peer review of "Comparative Analysis of Mannans Extraction Processes from Spent Yeast Saccharomyces cerevisiae"

_foods, 2022, doi:10.3390/foods11233753_

Round 1

Reviewer 1 Report

The paper has generally worked hard and with integrity.

1. A proper simplification of the introduction part is required.

2. Sections 2.5.2 and 2.6 of the experimental techniques can be appropriately simplified.

3. Why do the values for Mannose Purity (%) in Tab. 1 and Mannose (% w/w) in Tab. 3 match up?

4. The conclusion should be made simpler. It is a condensed version of the results and discussions rather than a straightforward repetition of the findings. 

Author Response

Journal: FOODS

Manuscript ID: foods-2015075

Previous title: Optimization of Mannans Extraction from spent yeast Saccharomyces cerevisiae

New title: Comparative analysis of Mannans extraction processes from spent yeast Saccharomyces cerevisiae

Authors: Margarida Faustino, Joana Durão *, Carla F. Pereira *, Ana Sofia Oliveira, Joana Odila Pereira, Ana M. Pereira, Carlos Ferreira, Manuela E. Pintado, Ana P. Carvalho

Dear Editor,

Please find enclosed the revised version of our manuscript foods-2015075 entitled “Comparative analysis of Mannans extraction processes from spent yeast Saccharomyces cerevisiae”.

The authors want to acknowledge the reviewers for the careful reading of the manuscript and all the comments and constructive suggestions made, which significantly contributed to improve the quality of the manuscript. All the suggestions and recommendations were taken into consideration and changes are highlighted using the “Track changes” function in Microsoft Word.

Reviewer 1

  1. A proper simplification of the introduction part is required. Thank you for the suggestion. Your suggestion has been considered and the alteration can be found in the new version of the manuscript.
  2. Sections 2.5.2 and 2.6 of the experimental techniques can be appropriately simplified. The experimental techniques were appropriately simplified as your suggestion.
  3. Why do the values for Mannose Purity (%) in Tab. 1 and Mannose (% w/w) in Tab. 3 match up? Thank you for the question. In fact, the values are the same. According to our understanding the mannose content is in fact the same as the mannose purity of the extracts as it reflects the content per mass. We harmonized the nomenclature to mannose content and eliminated this information from Table 3.
  4. The conclusion should be made simpler. It is a condensed version of the results and discussions rather than a straightforward repetition of the findings. We agree with the reviewers’ opinion and have changed the conclusion section accordingly.

Reviewer 2 Report

The manuscript entitled “Optimization of Mannans Extraction from spent yeast Saccharomyces cerevisiaerepresents the findings of the comparative study of different processes for the isolation of mannans from spent yeast S. cerevisiae. The study is interesting, and the design of experiments and writing deserve appreciation. However, the authors can consider the following minor suggestion to improve it further.

·       As the study mainly focuses on comparing the different processes for isolation of mannans, not on optimization, it will be better to reframe the title of the manuscript.

·       Authors should include a comparative table comparing the findings of other published works to the present study.

·       The authors concluded that the neutral pH thermal process is economically promising. The authors should conduct a cost analysis of the different processes to increase the value of this study.

Author Response

Journal: FOODS

Manuscript ID: foods-2015075

Previous title: Optimization of Mannans Extraction from spent yeast Saccharomyces cerevisiae

New title: Comparative analysis of Mannans extraction processes from spent yeast Saccharomyces cerevisiae

Authors: Margarida Faustino, Joana Durão *, Carla F. Pereira *, Ana Sofia Oliveira, Joana Odila Pereira, Ana M. Pereira, Carlos Ferreira, Manuela E. Pintado, Ana P. Carvalho

Dear Editor,

Please find enclosed the revised version of our manuscript foods-2015075 entitled “Comparative analysis of Mannans extraction processes from spent yeast Saccharomyces cerevisiae”.

The authors want to acknowledge the reviewers for the careful reading of the manuscript and all the comments and constructive suggestions made, which significantly contributed to improve the quality of the manuscript. All the suggestions and recommendations were taken into consideration and changes are highlighted using the “Track changes” function in Microsoft Word.

Reviewer 2

  • As the study mainly focuses on comparing the different processes for isolation of mannans, not on optimization, it will be better to reframe the title of the manuscript. Thank you for your suggestion. We modified the title of the manuscript for “Comparative analysis of Mannans extraction processes from spent yeast Saccharomyces cerevisiae”.
  • Authors should include a comparative table comparing the findings of other published works to the present study. Our team has previously published a review of mannans yeast extraction processes where we compiled this information in table format [20]. From the information retrieved from that review we were able to select the different processes conditions established for this study. In this particular paper we found it was enough to reference other teams’ work within the text when found necessary, as exemplified by the following quote of the paper:

 “The extraction of mannans usually includes the following steps: cell lysis, fractionation, and purification [20]. The step of cell lysis can be performed by different processes, namely by chemical (e.g., alkaline reagents such as NaOH [21,22] or buffer solutions such as sodium phosphate, citrate or Tris [6,7,23,24]), physical (using temperature and pressure), enzymatic (e.g., proteases [6,25,26], glucanases [6,27–29] and carbohydrases [7,30,31]), or mechanical processes (e.g., glass beads) [6]; a combination of processes is also widely employed. Their recovery from the solution (fractionation and/or purification) can be easily performed by precipitation – ethanol, methanol, or acetone are the solvents commonly reported in the literature for this purpose [24,32,33]. Additionally, chromatographic methods [31] and ultrafiltration [33,34] are also reported in the literature as methodologies to purify mannans.”

We considered that introducing the table could potentially cram the paper with information that was not necessarily the focus of the present paper.

  • The authors concluded that the neutral pH thermal process is economically promising. The authors should conduct a cost analysis of the different processes to increase the value of this study. Thank you for the suggestion. We were happy to follow your suggestion and found that it adds an interesting layer to the paper, by providing another fundamental perspective on the evaluation of the “best” performing process. We added a cost assessment section which considered two major cost factors – reagents and energy. A few assumptions were made and are described in the paper to allow the readers to perform their own cost assessments of the processes they perform, enabling the comparison with other works.

Reviewer 3 Report

Hi dear

This article "Optimization of Mannans Extraction from spent yeast Saccharomyces cerevisiae” was revised and has a novelty and I think it should consider the following comments.

Title: If you can rewrite and make it more interesting for readers. I propose: “Optimization of Mannan Extraction from spent yeast Saccharomyces cerevisiae”.

Abstract:

·       Line 15: FT-IR or FTIR?

·       Line 20: please point to the latter? As an obviously.

·       The type of statistical design used in this research should be mentioned.

·       Line 18 and Line 20: In the abstract and throughout the text of the article, the average number is sufficient and there is no need to state the standard deviation or standard error.

·       Please consider introduction as 3 to 4 paragraphs finally.

·       Line 92: why or? Please changed it to and.

·       Line 92: please express as a detail “ autolysis, and enzymatic hydrolysis”

·       Line 95: Please express as a detail “physicochemical properties”.

·       Please include as a detail and obviously treatments used.

·       Line 131-132: Yeast pellet (10% w/v) was suspended in NaOH solutions at 0.25 M 131

·       and 1.5 M, respectively. Please explain carefully which you provide two treatment.

·       Line 162-182 and Line 244-249: Some methods do not have references.

·       Line 185-192: Please assay the browning index, ΔE, Hue angle, chroma etc. and you can for the better understanding from “Food Science & Nutrition 9 (2), 866-874”.

·       All Tables: The alphabetical statistical letters for the means should all be modified such that the greatest number has the letter a and as the numbers go lower, letters b, c etc.

·       Table 2: Please at least provide browning index, ΔE index in Table 2.

·       Line 482-486 and so on: please delete standard error or standard deviation used in through the manuscript text.

·       Please provide an appropriate abbreviation for TH NaOH 1.5 M and the other treatments used throughout the manuscript text for reading convenience.

·       Line 633-639: it is not importance and I suggest delete it and provide the concise as a detail conclusion.

·       Discussion text must grammar improve and in some cases it is very weak and maybe there is no discussion at all.

The article has many flaws in express and concept of English, it is suggested to be revised in a scientific and native way.

Author Response

Journal: FOODS

Manuscript ID: foods-2015075

Previous title: Optimization of Mannans Extraction from spent yeast Saccharomyces cerevisiae

New title: Comparative analysis of Mannans extraction processes from spent yeast Saccharomyces cerevisiae

Authors: Margarida Faustino, Joana Durão *, Carla F. Pereira *, Ana Sofia Oliveira, Joana Odila Pereira, Ana M. Pereira, Carlos Ferreira, Manuela E. Pintado, Ana P. Carvalho

Dear Editor,

Please find enclosed the revised version of our manuscript foods-2015075 entitled “Comparative analysis of Mannans extraction processes from spent yeast Saccharomyces cerevisiae”.

The authors want to acknowledge the reviewers for the careful reading of the manuscript and all the comments and constructive suggestions made, which significantly contributed to improve the quality of the manuscript. All the suggestions and recommendations were taken into consideration and changes are highlighted using the “Track changes” function in Microsoft Word.

Reviewer 3

Title: If you can rewrite and make it more interesting for readers. I propose: “Optimization of Mannan Extraction from spent yeast Saccharomyces cerevisiae”. Thank you for your suggestion. We modified the title of the manuscript to “Comparative analysis of Mannans extraction processes from spent yeast Saccharomyces cerevisiae”. According to our understanding, the word mannans is more suitable to be used instead of mannan, because of the heterogeneity of the mannans extracts, particularly, on the molecular weight distribution analysis we found it to be multimodal.

Abstract:

  • Line 15: FT-IR or FTIR? In our understanding both forms are correct and can be found in several articles and are used interchangeably. However, we chose to use the abbreviation form accordingly with the manufacture’s manual (FT-IR; PerkinElmer, Frontier).  Indeed, in line 427 (with “Track changes”) the word was found without a hyphen, but this mistake has been corrected in the new version of the manuscript.
  • Line 20: please point to the latter? As an obviously. Thank you for the observation. We corrected this in the manuscript.
  • The type of statistical design used in this research should be mentioned. We are not sure if we correctly understood your question. In fact, we didn't use any statistical design for the different extraction methodologies tested, although a statistical analysis of the results was performed, according to the procedure described. Nevertheless, to enhance clarity, such procedure description was revised.
  • Line 18 and Line 20: In the abstract and throughout the text of the article, the average number is sufficient and there is no need to state the standard deviation or standard error. Thank you for the observation. We agree that it becomes a bit crowded, so we have modified accordingly in the new version of the manuscript.
  • Please consider introduction as 3 to 4 paragraphs finally. Your suggestion has been considered, and the alterations can be found in the new version of the manuscript.
  • Line 92: why or? Please changed it to and. It has been reformulated in the new version.
  • Line 92: please express as a detail “autolysis, and enzymatic hydrolysis”. We added information as required explaining which enzyme is used, however detailed information about the process can be found in the materials and methods section.
  • Line 95: Please express as a detail “physicochemical properties”. Thank you for the suggestion. We modified the text to clarify which were the methods included in physicochemical properties: colour, total protein, neutral sugars, molecular weight distribution, solubility, dry weight and ashes.
  • Please include as a detail and obviously treatments used.
  • Line 131-132: Yeast pellet (10% w/v) was suspended in NaOH solutions at 0.25 Mand 1.5 M, respectively. Please explain carefully which you provide two treatment. We are not sure if we correctly understood your question. In fact, the same methodology was used, but in one instance one NaOH 0.25M was used and in the other NaOH 1.5M was used. This results in two extracts with different characteristics.
  • Line 162-182 and Line 244-249: Some methods do not have references. Thank you for the comment. In fact, no references are presented as the methodology used by us is in accordance with the specifications/manuals of the different equipment
  • Line 185-192: Please assay the browning index, ΔE, Hue angle, chroma etc. and you can for the better understanding from “Food Science & Nutrition 9 (2), 866-874”. Thank you for the suggestion. We added the ΔE* of the different mannans extracts against a benchmark of mannans from S. cerevisiae by Sigma-Aldrich.
  • All Tables: The alphabetical statistical letters for the means should all be modified such that the greatest number has the letter aand as the numbers go lower, letters b, c etc. We corrected this topic in the new version of the manuscript.
  • Table 2: Please at least provide browning index, ΔE index in Table 2. We added the values of ΔE in Table 2.
  • Line 482-486 and so on: please delete standard error or standard deviation used in through the manuscript text. We corrected this topic in the manuscript. We removed the standard deviation from the majority of the text except in prebiotic assay because we considered it was important to keep it.
  • Please provide an appropriate abbreviation for TH NaOH 1.5 M and the other treatments used throughout the manuscript text for reading convenience. Although we understand that this abbreviation is slightly long, we also believe that it is important to clearly identify each one of the extracts obtained through their corresponding extraction methodology used. Therefore, and since there are two extracts obtained by thermal hydrolysis (TH) under alkaline conditions (NaOH), which only differ in their alkaline concentration (0.25 M and 1.5 M), we find it difficult to reduce the length of their corresponding acronyms. In the methods section we also added the abbreviation in the title of each process to facilitate.
  • Line 633-639: it is not importance and I suggest delete it and provide the concise as a detail conclusion. Your suggestion has been considered and the alteration can be found in the new version of the manuscript.
  • Discussion text must grammar improve and in some cases it is very weak and maybe there is no discussion at all. We recognize the reviewers’ comment and have improved the grammar and the discussion section. All the alterations can be found in the new version of the manuscript.

The article has many flaws in express and concept of English, it is suggested to be revised in a scientific and native way. We thank the reviewer for the suggestions and comments. We have performed a thorough review and hope to have improved the manuscript to an acceptable form for publication.
